# The Diagnostic and Prognostic Role of Biomarkers in Chronic Rhinosinusitis

**DOI:** 10.3390/diagnostics13040715

**Published:** 2023-02-14

**Authors:** Jun Jie Seah, Mark Thong, De Yun Wang

**Affiliations:** 1Department of Otolaryngology, Yong Loo Lin School of Medicine, National University of Singapore, Singapore 119228, Singapore; 2Department of Otolaryngology-Head and Neck Surgery, National University Hospital, National University Health System, Singapore 119228, Singapore; 3Infectious Diseases Translational Research Programme, Yong Loo Lin School of Medicine, National University of Singapore, Singapore 119228, Singapore

**Keywords:** biomarkers, diagnostic, prognostication, chronic rhinosinusitis

## Abstract

Chronic rhinosinusitis (CRS) refers to an inflammatory disease of the sinonasal mucosa, with a significant economic burden and impact on quality of life. The diagnosis of CRS is conventionally made on careful history and physical examination, including nasoendoscopic assessment which requires technical expertise. There has been increasing interest in using biomarkers in the non-invasive diagnosis and prognostication of CRS, tailored to the disease inflammatory endotype. Potential biomarkers currently being studied can be isolated from peripheral blood, exhaled nasal gases or nasal secretions, as well as sinonasal tissue. In particular, various biomarkers have revolutionized the way in which CRS is managed, revealing new inflammatory pathways where novel therapeutic drugs are employed to curb the inflammatory process, which may be different from one patient to the next. Biomarkers that have been extensively studied in CRS, such as eosinophil count, IgE, and IL-5, have been associated with a T_H_2 inflammatory endotype which correlates with an eosinophilic CRSwNP phenotype that predicts a poorer prognosis, tends to recur after conventional surgical treatment, but responds to glucocorticoid treatment. Newer biomarkers that demonstrate potential, such as nasal nitric oxide, can support a diagnosis of CRS with or without nasal polyps, especially when invasive tests such as nasoendoscopy are unavailable. Other biomarkers such as periostin can be used to monitor disease course after treatment of CRS. With a personalized treatment plan, the management of CRS can be individualized, optimizing treatment efficiency and reducing adverse outcomes. As such, this review aims to compile and summarize the existing literature regarding the utility of biomarkers in CRS in terms of diagnosis and prognostication, and also makes recommendations for further studies to fill current knowledge gaps.

## 1. Introduction to Chronic Rhinosinusitis (CRS)

Chronic rhinosinusitis (CRS) refers to an inflammatory process of the sinonasal mucosa that lasts for more than 12 weeks without resolution of symptoms. It represents a global health burden with a widespread prevalence, affecting about 5–12% of the general population [1]. CRS is estimated to cost approximately USD $30 to $33 billion per year in terms of direct and indirect costs [2], posing a significant economic burden and detriment to quality of life. CRS is traditionally categorized as CRS with nasal polyps (CRSwNP) or CRS without nasal polyps (CRSsNP). CRSwNP can be further subdivided into eosinophilic CRSwNP (eosCRSwNP) or non-eosinophilic CRSwNP (non-eosCRSwNP) depending on the extent of eosinophilic inflammation taking place. Previous studies have demonstrated that certain clinical phenotypes and endotypes of CRS are geographically clustered [3,4], thereby influencing clinical practice and treatment options. Newer studies have however revolutionized this oversimplistic classification, with newer findings suggesting that CRS comprises a heterogenous group of diseases that span a wide spectrum of related conditions. There are also an increasing number of studies focused on CRS endotyping via characterization of molecular pathways of inflammation.

The diagnosis of CRS foremostly involves a comprehensive history and physical examination including nasoendoscopy, supported by imaging features of mucosal inflammation. Subsequent treatment of CRS may include pharmacological therapy such as nasal sprays or douches, systemic antibiotics and steroids, immunomodulators, as well as surgical intervention such as endoscopic sinus surgery (ESS). Yet after treatment of CRS, many patients can have disease recurrence or inadequate relief. It is not easy to tell which patients will be successfully treated as opposed to patients who are predisposed to recurrent or more aggressive disease. Moreover, not all patients, especially those living in rural areas, have easy access to tertiary care where serial nasoendoscopic assessment is readily available to chart disease course after treatment with medical or surgical therapy. There is thus a potential role for biomarkers to tackle these gaps.

## 2. Biomarkers in CRS

Biomarkers refer to particular substances that can be detected and measured from a specimen obtained from the human body and that can provide information on disease state. Every so often, new biomarkers are discovered, which rapidly evolves the way in which diseases are being diagnosed and managed. Biomarkers confer many advantages in the diagnosis, management, and prognosis of diseases, providing a non-invasive surrogate of the underlying disease process which would otherwise have been difficult or sometimes unfeasible to determine. Certain biomarkers can also provide an early indicator of undesirable outcomes, which may not have manifested clinically yet, allowing the clinician to anticipate the disease early in its course and prepare for optimization of subsequent management plans. Last but not least, biomarkers, when identified as part of a larger inflammatory pathway, can present novel therapeutic targets that may augment or replace traditional treatment modalities.

In the context of CRS, common sources of biomarkers include blood samples obtained from a peripheral venepuncture, nasal secretions obtained via nasal lavage, as well as gaseous substances or molecules obtained from exhaled nasal breath. If surgery or a biopsy has been performed (e.g., nasal mucosal tissue, nasal polyps, etc.), tissue specimens can be processed and further assayed for levels of biomarkers. These signify a diverse range of specimens that can be obtained for biomarker quantification, suggesting that the realm of biomarker detection in CRS is boundless. This review hence aims to explore the current literature pertaining to the utility of current biomarkers in the diagnosis and prognosis of CRS and its various subtypes.

In terms of the therapeutic realm, research on novel therapies has been garnering increasing interest, especially when it comes to the uses of biologic therapy such as monoclonal antibodies targeting components of type 2 inflammation in CRS. While such therapeutic options usually act through chemical intermediaries such as interleukins or other cytokines, eosinophils act as the eventual effector cell in the inflammatory pathway and can often be the first clue of a type 2 inflammatory disease endotype. Biomarkers can potentially bridge this link by providing a means of monitoring treatment response, assess for efficacy of novel biologic therapy, as well as provide a quantitative means to assess disease severity. Such treatment modalities are subsequently explored in this review in the respective sections below, but it is worthwhile to note that therapeutics, including novel biologics, have multiple downstream effects given the vast interplay amidst the milieu of type 2 inflammation. This review hence also explores the interplay between biomarkers in CRS and studies the ways in which biomarkers support the development of novel treatment modalities such as biologic treatment.

## 3. Potential Biomarkers in the Diagnosis and Prognosis of CRS

### 3.1. Eosinophil Count

Eosinophils are a subtype of leukocytes which belong to the granulocyte subcategory, and contain many granules containing inflammatory chemical mediators such as major basic protein (MBP), eosinophil cationic protein (ECP), eosinophil peroxidase (EPX), and eosinophil-derived neurotoxin (EDN) [5]. They function mainly in the mediation of allergic disease, parasitic as well as helminthic infections, and participate in type 2 (T_H_2) inflammation that involves key cytokines such as IL-4, IL-5, and IL-13 [6]. Eosinophils can be measured in peripheral blood as well as in nasal tissue specimens, which is widely available in almost all clinical settings. Normal levels of peripheral blood eosinophils range from 0 to 0.5 × 10^9^ cells per liter [7].

CRS, being a disease where allergy plays a contributing role, features eosinophilic inflammation to varying degrees, especially in CRSwNP [8]. However, several studies conducted in different parts of Asia found that over half of CRSwNP cases in Asian patients present with noneosinophilic inflammation [9,10]. Given the heterogeneity and disagreement in the current literature, many studies have investigated the utility of blood eosinophils in the workup, monitoring, and prognostication of CRS.

Peripheral blood eosinophil count is helpful in supporting a diagnosis of CRS, in particular the eosinophilic subtype. Sakuma et al. investigated 124 patients with CRSwNP who underwent ESS, and found that a blood eosinophil percentage greater than 6% predicted eosCRSwNP with a sensitivity and specificity of 97.4% and 70.7%, respectively, with improved diagnostic accuracy if combined with computed tomography scan (CT) findings [11]. Zuo et al. studied both CRSwNP and CRSsNP patients, and found that an absolute blood eosinophil count of 0.16 × 10^9^ cells per liter had a sensitivity and specificity 84.9% and 84.4%, respectively, for diagnosing eosCRS [12]. The use of peripheral blood eosinophilia is however limited, because it is non-specific for CRS and can be elevated in a myriad of other diseases, ranging from parasitic infection, allergic or autoimmune diseases, even to medications [13]. Tissue eosinophilia may also portray a more accurate representation of local inflammatory activity, although peripheral blood eosinophilia has been identified as a surrogate marker for tissue eosinophilia in both diagnostic and prognostic studies [12,14].

In terms of prognostication, Ho et al. discovered in their multivariate logistic regression model that a peripheral blood eosinophil count of more than 0.455 × 10^9^ cells per liter is associated with approximately nine times increased likelihood of requiring long term systemic medication following ESS in eosCRS, with a high negative predictive value of 98.7% [15]. Interestingly, a study by Honma et al. found that in CRS patients, high pre-ESS peripheral blood eosinophil count that decreased significantly post-ESS was associated with a higher likelihood of suffering from disease recurrence [16]. These studies are in general agreement with the literature which suggests that eosinophilic inflammation in CRS is associated with poorer prognosis.

Tissue eosinophilia, on the other hand, measures the proportion of eosinophils in a sample of mucosal tissue. As mentioned above, tissue eosinophilia may better reflect local eosinophilic inflammatory activity. Relying on tissue eosinophilia, however, requires the patient to undergo invasive procedures such as a biopsy or surgical resection, which is not without attendant risk. It can however provide valuable information that can guide subsequent management and treatment optimization, especially if the patient is already undergoing surgical therapy.

Eosinophilia in nasal polyp specimens is associated with an increase in objective disease severity in terms of endoscopic and radiologic scores [17]. Wen et al. found that eosinophilic polyps were more glucocorticoid responsive than noneosinophilic polyps [18], while in other studies noneosinophilic or neutrophilic CRSwNP are resistant to steroid treatment [19,20]. It was also found that increased levels of eosinophils within the olfactory tissue were strongly related to olfactory loss in CRSwNP patients [21], implying that local eosinophils in tissue might act as essential effectors in olfactory dysfunction. It is hence helpful to identify the subset of patients with eosinophilic nasal polyps as it can predict disease course and influence patient selection for therapeutic modalities tailored to the disease endotype.

A myriad of studies have attempted to determine a suitable cut-off point to define tissue eosinophilia. While these studies unanimously show that a higher proportion of tissue eosinophilia is associated with poorer outcomes, there is no agreement on a standardized cut-off value that can be recommended to clinicians for everyday use. Kountakis et al. proposed a cut-off level of greater than five eosinophils per high power field (HPF) for sinus tissue eosinophilia, and these patients had more severe disease as determined by CT and endoscopy scores [22]. Soler et al. defined a higher cut-off where more than 10 eosinophils per HPF was considered mucosal eosinophilia, which was able to predict reduced improvement of quality of life after ESS [23]. Lou et al. studied a cohort of 387 Chinese patients with CRSwNP, following them up for at least 24 months, and demonstrated that a tissue eosinophil proportion of greater than 27% or an absolute tissue eosinophil count of greater than 55 eosinophils per HPF predicted the relapse of nasal polyps within 2 years after ESS [24]. Tokunaga et al. investigated a larger cohort of 1716 patients in Japan undergoing ESS for CRS, and found that tissue mucosal eosinophilia of more than 70 eosinophils per HPF was strongly associated with recurrence after surgery with the greatest statistical significance [25]. With regards to eosCRSwNP, McHugh et al. attempted to summarize the existing literature in their meta-analysis, and found that an eosinophil count greater than 55 per HPF was associated with disease recurrence and treatment failure following ESS [26].

As can be seen, the existing medical literature pool boasts a wide amount of information regarding the utility of eosinophil count, both in blood and tissue specimens, in the diagnosis and prognostication of CRS. Its utility is mainly limited by its lack of specificity, given that it may be raised in many other inflammatory conditions apart from CRS.

### 3.2. Immunoglobulin E (IgE)

IgE is an antibody that is produced by plasma cells and is involved in allergic, eosinophilic, or parasitic diseases. There is a growing repertoire of evidence to show that IgE, whether measured in peripheral blood, nasal secretions, or tissue specimens, may have an important role in the diagnosis and prognostication of CRS.

#### 3.2.1. Serum IgE

IgE can be similarly quantified from a venepuncture sample and provide information regarding the underlying disease process. Wei et al. studied 78 CRSwNP patients who underwent ESS and followed them up for a period of eight years, and discovered that total serum IgE was significantly elevated in recurrent CRS when compared with their non-recurrent counterparts [27]. A study by Ho et al., however, seemed to disagree and found no significant association between total serum or serum specific IgE and eosCRS [28]. Mori et al. found that serum total IgE level of greater than 400 IU/mL was identified as an independent risk factor for olfactory dysfunction in eosCRS patients [29]. This association however did not hold true when analyzing all types of CRS as a whole. Gevaert et al. found that treatment with anti-IgE agents such as omalizumab could contribute to a significant improvement in subjective olfactory function [30], suggesting a potential therapeutic option for patients with such symptoms.

#### 3.2.2. Tissue IgE

A Japanese study by Baba et al. investigated 44 CRS and control patients and discovered that tissue IgE concentrations were significantly higher in eosCRS polyps as compared to non-CRS controls [31]. In a prospective follow-up study performed over 12 years after ESS, Gevaert et al. showed that high local IgE in tissue itself was also predictive of disease recurrence requiring repeat surgical intervention in CRSwNP patients [32]. A randomized double-blinded study of CRSwNP patients with comorbid asthma also found that omalizumab, a recombinant humanized monoclonal antibody targeting IgE, reduced nasal polyp size and improved sinus CT scores in patients with CRSwNP. Unfortunately, decreases in local nasal mucosal inflammation were not unequivocally observed [30].

Local mucosal tissue polyclonal IgE is a major contributing factor in the disease process of CRSwNP, with studies showing promising results [33]. Bachert et al. found that total IgE concentration in NPs correlated significantly with levels of ECP and IL-5, and by extension local eosinophilic inflammation [34]. Furthermore, in patients with CRSwNP, tissue IgE and specific IgE against *Staphylococcus aureus* enterotoxins were identified as risk factors for the comorbidity of asthma [35].

Identifying patients with raised IgE can prove beneficial as it opens up therapeutic options for patients with refractory disease. In a recent proof-of-concept clinical trial, omalizumab demonstrated significant improvements in clinical parameters, symptoms, and QOL measures in patients with CRSwNP and comorbid asthma, irrespective of their allergic status [30]. Indeed, newer agents are continuously being discovered and adds to the armamentarium of therapeutic options available to the otolaryngologist, and may be an attractive therapeutic target if patients are carefully selected. Cost and side effect profile are the main drawbacks when considering the use of biologics in treatment [36,37].

#### 3.2.3. Nasal Secretions IgE

IgE can be isolated from nasal secretions, providing a non-invasive means of quantifying the extent of eosinophilic inflammation, and by extension, CRS. In a study by Riechelmann et al. involving 38 patients with rhinosinusitis and controls, it was found using principal component analysis that high nasal IgE as well as high nasal IL-5 levels were specific for CRSwNP [38]. Dupilumab, which is a monoclonal antibody targeting the IL-4/IL-13 receptor, has been shown in a phase 3 trial involving 448 patients with CRSwNP to reduce IgE in nasal secretions, as well as serum total IgE [39].

### 3.3. Eosinophil Granule Proteins

#### 3.3.1. Eosinophilic Cationic Protein (ECP)

ECP, also known as ribonuclease 3, is a protein with cytotoxic properties produced and stored within eosinophilic granules. It is released during eosinophilic degranulation in response to eosinophilic activation and inflammation [40]. It has previously been shown that ECP is raised in diseases such as asthma [41] and atopic dermatitis [42]. The normal reference range for blood tests for ECP is between 2.3 and 16 micrograms per liter [43].

Van Zele et al. previously showed that serum ECP is raised in patients with CRSwNP [44]. Tsuda et al. showed that serum ECP was significantly elevated in patients with eosCRS, but unlike serum EDN (see below), serum ECP was not associated with clinical markers of severity [45]. Another study by Kim et al. retrospectively analyzed 492 patients with CRS and/or allergic rhinitis who underwent ESS or septoplasty, and showed that elevated serum ECP correlated with raised blood eosinophils and increased eosinophil concentration on analysis of nasal smears [46]. ECP has also been measured in nasal secretions and shown to correlate with blood eosinophil levels [47]. Overall, a vast majority of studies seem to suggest that there may be utility in quantifying ECP levels especially in the eosinophilic subtype of CRS, understandably so given its relationship to other diseases that feature a similar inflammatory signature such as asthma and atopic dermatitis.

#### 3.3.2. Eosinophil-Derived Neurotoxin (EDN)

EDN is another eosinophil degranulation product that has been previously shown to be a novel biomarker with clinical utility in asthma [48]. It has cytotoxic properties and plays an important role in immunity in terms of attracting and enhancing the activity of antigen-presenting cells, and activation of immune cells via Toll-like receptor 2 [49]. Tsuda et al. studied 115 patients with nasal and paranasal disease and found that serum EDN is significantly elevated in patients with eosCRS as compared to patients with non-eosinophilic sinus disease, as well as healthy disease-free controls. They also found that serum EDN correlated with clinical indicators of disease severity such as disease activity, polyp score, and peripheral eosinophilia [45]. They hypothesized based on their study findings that when stimulated by IL-5, EDN degranulation promotes MMP-9 secretion from the nasal epithelium, possibly accounting for nasal tissue remodelling and subsequently nasal polyposis. While studies have attempted to characterize the effects EDN, the concept of using EDN as a biomarker in CRS is still relatively novel, and more studies are required to investigate in further detail its practical utility in everyday practice.

#### 3.3.3. Eosinophil Peroxidase (EPX)

EPX is an enzyme found within the secretory granules of eosinophils and is involved in the formation of bactericidal reactive oxygen species [50]. Wang et al. previously found that EPX is implicated in the pathogenesis of asthma [51]. With respect to CRS, Lal et al. retrospectively studied 93 patients who underwent sinonasal surgery and found that tissue EPX staining was higher in CRS tissue compared to controls [52]. Tsuda et al. however investigated the relationship between serum EPX and eosCRS, and found no association [45]. At present, literature pertaining to EPX seems to be in disagreement and will benefit from more studies to further elucidate its clinical utility.

### 3.4. Periostin

Periostin is an matricellular protein involved in the pathologic remodelling processes seen in eosinophilic inflammation. It is produced by epithelial cells when stimulated by IL-4 and IL-13 [53], and also acts as a marker in the aforementioned remodelling process [54].

In terms of diagnostic utility, periostin levels show promise in identifying patients with CRSwNP. Wang et al. confirmed that periostin expression is increased in patients with CRSwNP, independent of their atopic status or comorbid asthma [55]. Jonstam et al. found that serum periostin levels were significantly elevated in CRSwNP compared to CRSsNP and controls [56]. Another study by Ohta et al. showed that elevated periostin in serum and nasal fluids could distinguish between patients with CRSwNP against healthy subjects and patients with allergic rhinitis [57]. A recent study comprising 94 patients with CRSwNP revealed that serum periostin is significantly elevated in patients with eosinophilic polyps compared to those with non-eosinophilic polyps [58]. They found that an optimal cut-off level for serum periostin of 83.41 ng/mL based on ROC analysis yielded a sensitivity of 72.9% and a specificity of 60.9% for the diagnosis of eosCRSwNP.

In terms of prognostication, periostin levels can also non-invasively clue the clinician in towards the severity of disease. This is demonstrated by Ninomiya et al., who found that serum periostin was significantly elevated in patients with severe eosCRS based on the JESREC scoring [25], as compared to patients with mild eosCRS or without eosCRS. They also found that high serum periostin above a cut-off of 115.5 ng/mL significantly predicted postoperative recurrence in CRS [59]. Going further, tissue periostin expression also correlates with disease activity. Zhang et al. showed that periostin levels from frontal recess biopsy samples, quantified via immunohistochemistry staining, were elevated in active CRS preoperatively and subsequently decreased after disease resolution with ESS [60]. Kim et al. also found that periostin mRNA and protein levels were significantly increased in nasal polyp tissue compared to uncinate process tissue. Periostin concentrations in nasal polyp homogenates of eosCRSwNP patients was also found to correlate positively with Lund-Mackay CT scoring [61]. These studies suggest that periostin may be a potential biomarker to assess and track disease course, above and beyond its role in diagnosing CRS.

In patients with eosCRS, there was a significant positive correlation between serum periostin levels and IL-5 expression [55]. Jonstam et al. performed ROC analyses on 108 CRSwNP patients and found that the best cut-off level for periostin to predict the presence of tissue IL-5 was 48.5 ng/mL, with a sensitivity and specificity of 93.5% and 62.5%, respectively [56]. Tissue periostin levels were significantly associated with IL-5 expression [55]. In eosCRSwNP, tissue periostin levels were also found to correlate positively with radiographic findings [61].

While there are no proven therapies against periostin in CRS, Tomaru et al. studied the effect of intranasal siRNA and antisense oligonucleotides against periostin in the context of idiopathic pulmonary fibrosis. Their results showed significant reduction in lung tissue periostin levels, collagen deposition, and lung fibrosis score [62], showing promise for future potential CRS treatment modalities.

### 3.5. Interleukins

Interleukins represent a diverse group of chemical molecules that function in cell signalling. Depending on the type of inflammation taking place, different interleukins are found in greater concentrations than others, allowing for an “inflammatory signature” to be characterized in different disease subtypes and in different patients. In terms of diagnostic function, interleukins may play a role in supporting the diagnosis of chronic rhinosinusitis.

#### 3.5.1. Type 2 Interleukins (IL-4, IL-5, IL-13)

Upregulation of IL-5 mRNA expression within tissue has been associated with eosCRS [31,63,64]. Tomassen et al. performed a multicenter case-control study involving 682 subjects with CRS that clustered patients by immune markers solely, so as to correlate said clusters to clinical CRS phenotype. They found that negative, moderate, and high IL-5 clusters were associated with clinical phenotypes of CRSsNP, mixed CRSsNP/CRSwNP, and an almost exclusively nasal polyp phenotype, respectively [65].

In terms of prognostication, Gevaert et al. performed a long-term prospective cohort study involving 47 patients over 12 years and found that local IL-5 and IgE production were risk factors for requiring revision surgery after ESS in patients with CRSwNP [32]. Van Zele et al. also performed a similar study which found that tissue IL-5, and in addition specific IgE to *Staphylococcus aureus* enterotoxin and ECP, were significantly increased in recurrent versus nonrecurrent CRSwNPs [66]. In the same study, they also discovered that interferon gamma protein levels were significantly higher in nonrecurrent CRSwNPs. Schlosser et al. compared inflammatory mediators found in mucus from the olfactory cleft against objective olfactory function as assessed via the Sniffin’ Sticks test and found that IL-5 levels correlated inversely with olfactory scores irrespective of polyp status [67]. IL-5 as well as IL-6 mRNA expression has been shown to be significantly elevated in the frontal sinus mucosa compared with ethmoid sinus mucosa within the same eosCRS patient, which may be a factor contributing to recurrence [64]. Bachert et al. compared nasal polyps from Belgian and Chinese patients, owing to their differences in prevalence of asthma as a comorbid disease. They confirmed that tissue IL-5 was the main positive determinant of eosinophilic inflammation and that IL-5 positive nasal polyps were associated with a higher likelihood of suffering from comorbid asthma [35]. This is unsurprising, given that IL-5 plays an important role in eosinophil recruitment, activation, and survival [68]. Other studies have also shown that increased expression of IL-5 at both the mRNA and protein levels has been identified in nasal polyps compared with sinus mucosa, and is associated with the need for revision surgery and nasal polyposis [66,69].

The role of IL-5 in CRS is further exemplified in clinical studies demonstrating potential utility in IL-5 antagonism in CRSwNP. Studies performed involving the administration of humanized monoclonal antibodies against IL-5, namely reslizumab and mepolizumab, found that there was significant reduction in the size of nasal polyps as well as peripheral blood eosinophil level [70,71], which reduced the necessity for revision surgeries [72]. Eosinophil markers were however not predictive of response to mepolizumab [73]. Tomassen et al. also showed that the levels of IL-5 in nasal secretions could predict treatment response to reslizumab [65]. It can be seen that biomarkers not only help with prognostication in CRS, but also understanding their role in the inflammatory pathway can reveal valuable insights into potential therapeutic targets, such as those conferred by novel therapeutic options such as monoclonal antibodies.

IL-4 and IL-13 are essential cytokines with shared signalling pathways and function in the activation of type 2 inflammatory responses [74]. In recent years, there has been increasing interest in targeting the IL-4/13 signalling pathway. Turner et al. investigated the levels of T_H_2 cytokines (namely IL-4/5/13) in nasal secretions and found that elevated IL-5 and IL-13 were not only found in CRS when compared to healthy controls, they were associated with worse objective measures of disease severity and greater rates of revision surgery. Their findings held true even when CRSwNP patients were analyzed separately [75]. A recent Phase II clinical trial by Bachert et al. showed that dupilumab, a fully human monoclonal antibody targeting the α subunit shared by IL-4 and IL-13 receptors, significantly improved clinical, endoscopic, radiological, and pharmacodynamic outcomes in patients with nasal polyposis after 16 weeks [76]. On the other hand, it has also been recognized that there is a lack of association between IL-4/13 and conventional measures of disease burden [54].

#### 3.5.2. Other Interleukins

Apart from the conventional interleukins involved in type 2 inflammation, a whole host of other interleukins have been studied in the context of CRS diagnosis and prognostication. A study by Lackner et al. showed that IL-16 is significantly raised in patients with eosCRS as compared to healthy controls [77]. Lam et al. showed that IL-25 expression was not only upregulated in CRSwNP tissue but was also associated with elevated eosinophil counts [78] and increased severity in CT scoring parameters as shown in a subsequent study [79]. IL-25 can hence potentially serve as a sensitive biomarker [80]. In a murine model, Shin et al. showed that anti-IL-25 treatment reduced the number of nasal polyps, mucosal edema thickness, collagen deposition, and infiltration of inflammatory cells, such as eosinophils and neutrophils [81], suggesting that IL-25 may serve as a novel pharmacologic target for CRSwNP patients.

IL-33 also plays a role in T_H_2 inflammation, but is not as extensively studied as other conventional cytokines. Ozturan et al. found that mean tissue IL-33 levels in CRSwNP was statistically significantly lower than CRSsNP and healthy control patients, and that IL-33 levels negatively correlated with Lund-Mackay CT scores [82]. Another study by Reh et al. discovered that treatment-recalcitrant CRSwNP epithelial cells had increased baseline expression of IL-33 compared to treatment-responsive CRSwNP [83], potentially serving as a predictor for treatment response, whether the treatment instituted was medical or surgical. A study by Kim et al. on Asian subjects showed that not only was IL-33 mRNA and protein increased in patients with CRSwNP when compared to controls, IL-33 protein concentration correlated positively with various T_H_1/T_H_17 cytokines and neutrophil recruitment [84]. They went further to investigate the effect of an IL-33 neutralizing monoclonal antibody in a murine model and found that anti-IL-33 treatment diminished mucosal thickness of edematous mucosa and subepithelial collagen deposition, and inhibited infiltration of neutrophils but not eosinophils.

Oncostatin M, a cytokine related to the IL-6 family, functions in signal transduction and has effects in mediating inflammatory processes [85]. Its exact mode of action and effects are still under constant investigation and are presently unclear. Nonetheless, it has been found to be elevated in CRSwNP and causes reversible reduction in barrier function of in vitro sinonasal epithelial cells [86].

### 3.6. Eotaxin-3

Eotaxin-3 is a cytokine produced by epithelial cells which induces chemotaxis of eosinophils, basophils, and T_H_2 lymphocytes [87]. Bachert et al. found that tissue eotaxin concentrations in nasal polyp specimens were significantly higher as compared to nonpolyp tissue [34]. Ikeda et al. also performed a cross-sectional study on Japanese patients with CRSwNP and classified patients by the extent of inflammatory cell infiltrate. They found that the expression of eotaxin in nasal polyps was greater in the eosinophilic group compared to the neutrophilic and non-eosinophilic non-neutrophilic group, and is also associated with poorer prognosis [88]. Plasma levels of eotaxin-3 have also been shown to be significantly elevated in patients with tissue eosinophils greater than 55 per HPF and may be useful as a marker for disease recurrence [89]. The phase 3 trial by Bachert et al. (mentioned earlier under *Tissue IgE*) also found that in CRSwNP patients, treatment with dupilumab reduced levels of plasma eotaxin-3 [39]. While studies into biomarkers such as eotaxin in CRS may still be evolving, these suggest that eotaxin may be a potential therapeutic target in the treatment of CRS in the future.

### 3.7. Nasal Nitric Oxide

Nitric oxide (NO) is an endogenous mediator produced via the activity of nitric oxide synthase (NOS). NOS is present in either constitutive or inducible forms, of which the latter is involved in inflammatory processes [90]. NO is constantly produced by different areas of the respiratory epithelium and exhaled into the external environment [91].

Nasal NO levels (nNO) are decreased in patients with nasal polyposis, and studies have attributed this phenomenon to nasal polyps resulting in sinus ostial occlusion, preventing NO within the sinus cavities from being excreted via the nasal cavities into the external environment [92,93]. Liu et al. corroborated this finding in their study which showed that patients with CRSsNP also have lower nNO than control patients on average as well, but significantly higher than those observed in CRSwNP [94]. A possible explanation to account for this may be due to mucosal edema and impaired mucociliary clearance leading to sinonasal obstruction, albeit not reaching the level of obstruction when nasal polyps are present. Several studies have attempted to define a cut-off value for nNO for the diagnosis of CRSwNP. Bommarito et al. showed in a study of 44 patients with CRS that a nNO cut-off value of 442 ppb could screen for and diagnose patients with CRSwNP with a sensitivity and specificity of 87% and 91%, respectively [95]. Jeong et al. studied 89 patients with CRSwNP, allergic rhinitis, and healthy controls, and found that a cut-off exhaled nNO of 163 ppb predicted CRSwNP with a sensitivity and specificity of 81.3% and 93.3%, respectively [96]. There may be a possible role for using nNO in the detection of CRS in patients without active or overt symptoms, or when nasoendoscopy is not available, such as in the primary care setting. This could also potentially facilitate accurate patient selection for those who may benefit from referral onward for specialist evaluation.

Studies have also suggested a possible role of nNO to be an objective means of quantifying the severity of CRSwNP non-invasively. A few studies have shown that in patients with CRSwNP, initial nNO levels correlated inversely with the extent of sinus disease, quantified by nasoendoscopy findings and CT changes [92,93,95,96,97]. After treatment of CRS, whether medically or surgically, studies are in general agreement that nNO increased in parallel with clinical markers of disease severity [93,97,98]. Deroee et al. investigated levels of NO metabolites in nasal lavages in CRS patients, and they too found that, similar to nNO, levels increased to normal after ESS along with subjective improvement of symptoms [99].

### 3.8. Matrix Metalloproteinases (MMPs)

MMPs are protease enzymes that participate in extracellular matrix (ECM) degradation [100]. There are various subtypes of MMPs, of which 23 have been identified in humans. In particular, MMP-9 levels have been extensively reported to be increased in CRSwNP [101,102,103,104], which may lend support to the diagnosis of CRS.

In terms of prognostication, Watelet et al. found that the concentration of MMP-9 in nasal fluid correlated with MMP-9 expression in the ECM of nasal tissue samples, which was in turn linked to poor healing quality [105]. They went further to perform another prospective study on 36 patients undergoing functional ESS for CRS or nasal polyposis, and found that nasal fluid MMP-9 can predict healing outcome after surgery and that high concentrations of MMP-9 post-operatively were associated with poor healing [106], suggesting a potential role for quantifying MMP-9 in the prognostication of CRS. On the contrary, no significant correlation has been demonstrated between MMP-9 expression and the severity of CRSwNP assessed by CT scans and polyp grades [102,104].

The role of MMP in CRS is further supported by several studies that investigated the relationship between doxycycline and MMP-9. A double-blinded prospective study by Van Zele et al. investigated the effect of doxycycline on 47 participants with CRSwNP and showed that doxycycline significantly lowered nasal secretion of MMP-9, ECP, and myeloperoxidase, as well as decreased polyp size with moderate effect for up to 12 weeks [107]. Another study by Huvenne et al. developed doxycycline-releasing stents delivered locally to the frontal sinus and found that this decreased MMP-9 concentrations compared to placebo stents, with associated improvement in postoperative healing quality [108]. Doxycycline is known to possess an anti-MMP effect that lowers MMP levels in secretions and causes decreases in polyp size [109], and its effects on CRS patients may suggest an integral role of MMP in the pathophysiology of CRS.

### 3.9. P-glycoprotein

P-glycoprotein, also known as multidrug resistance protein 1, is an ATP-dependent efflux protein pump present in cell membranes responsible for pumping many foreign substances out of cells. It is upregulated in T_H_2 disease [53] including CRSwNP [110,111]. Elevated p-glycoprotein levels are seen in all CRS subtypes, with higher levels in CRSwNP, and confer worse subjective and objective measures of disease severity [112]. Verapamil is a first-generation antagonist of p-glycoprotein, and preliminary trials show that low-dose verapamil therapy is safe and effective for CRSwNP treatment. Improvements in SNOT-22 score with verapamil were comparable with those achieved with steroids or biologic agents [113].

### 3.10. Glucocorticoid Receptor β (GR_β_) 

GR_β_, an endogenous inhibitor of glucocorticoid action, appears to be linked to steroid treatment response in CRSwNP [114]. In one study, Hamilos et al. reported an inverse relationship between baseline GR_β_ expression in nasal polyp inflammatory cells and the efficacy of topical fluticasone [115]. A study by Valera et al. concurs, with similar findings of higher GR_β_ expression in poor responders following treatment with intranasal budesonide for two months [116]. It appears that quantifying GR_β_ may be a potential means to predict response to topical steroid treatment, which may influence subsequent treatment plans to optimize treatment response and reduce recurrence rates.

### 3.11. Mucins

Mucins belong to a family of glycoproteins expressed on the surface of many epithelial cells. While mucins share common structural features, subtypes differ based on their tandem repeat peptides and are distinct on this basis [117]. As a whole, mucins have important functions in cell signalling, immunoregulation, and inhibition of cell adhesion [118]. Given that CRS is an inflammatory disease with intricate involvement of the immune system, it is unsurprising that mucins may be of value in the diagnosis and prognostication of CRS.

Studies on the anti-inflammatory effects of glucocorticoids demonstrated that the abnormal expression level of mucin 1 (MUC1) and mucin 4 (MUC4) might be functionally involved in steroid resistance. Milara et al. showed that MUC1 expression was significantly downregulated in the nasal polyp epithelium of CRSwNP patients resistant to systemic corticosteroids [119]. Opposite results were however observed for MUC4 expression [120].

### 3.12. Taste Receptors

There has also been an increasing role for genetic biomarkers to be utilized in the prognostication of CRS. In a study by Adappa et al., they found that patients with a nonfunctional polymorphism in a specific taste receptor, T2R38, have inferior outcomes after ESS and require increased intervention [121]. Another study found that sinonasal specimens from patients with nonfunctional polymorphisms exhibit increased bacterial biofilm formation in vitro [122].

Patients with CRSsNP are significantly more likely to be less sensitive to denatonium, a broad, bitter taste receptor agonist, and patients with CRSwNP and CRSsNP are more sensitive to sucrose, which is a sweet receptor agonist [123]. This potentially suggests that using genetic biomarkers to identify such taste receptor polymorphisms in CRS patients may reveal novel adjunctive therapeutic options. More studies are however needed as there is still a paucity of literature pertaining to this potential biomarker.

## 4. Future Needs

While this review has attempted to consolidate and synthesize the current literature pertaining to the utility of biomarkers in the diagnosis and prognostication of CRS, there remains a plethora of information that has yet to be uncovered. Table 1 summarizes what this review has explored and provides an overview of potential biomarkers at-a-glance. Existing knowledge gaps are also highlighted for future studies.

Many studies described in this review mainly show the association between a certain biomarker and the state of disease in CRS. There are not many studies, however, focusing on elucidating the pathophysiologic mechanisms by which the biomarker of interest is influenced by the presence of CRS disease, disease activity, subtype, or even prognosis. If more studies are performed to delineate such pathways, these new information could reveal deeper insights into the complexity of CRS as a disease. Figure 1 consolidates the various biomarkers explored in this review and their interaction in the inflammatory pathway leading to CRS. Potential treatment modalities are also included for a better overview.

Regarding certain specific biomarkers, there still exist disagreements between studies, suggesting that there may still be a long way to go for such biomarkers before they can be implemented in the clinical setting. Further studies could explore in further detail such biomarkers to see how they can be harnessed in clinical practice with better efficacy. Other biomarkers which have been more extensively studied may benefit from standardization of measurement and interpretation, such that they may be universally translated into clinical use in a practical manner. Such further studies may take the form of prospective case-control studies, which can better illustrate if a particular biomarker can be applied in clinical practice.

In terms of therapeutics, there has been increasing research interest in studying the utility of biologic treatment in CRS, which shows promising results thus far. A recent review by Xu et al. comprehensively summarizes the current biologic therapies available in CRSwNP and NSAID-exacerbated respiratory disease, and proposed a four-step approach to biologic selection [124]. If future studies can increase the emphasis on incorporating more biomarker data into biologic selection, they can potentially serve as another viewpoint in which the disease course of CRS is assessed and monitored.

Regardless of the current information deficit, researchers have come a long way in expanding our current understanding of biomarkers in CRS. It is only with continuous efforts to push the current frontiers of this topic that we will hopefully be able to channel it towards better patient outcomes.

## 5. Conclusions

In conclusion, the role of biomarkers in the diagnosis and prognostication of CRS remains a relatively unexplored field, yet the abundance of studies done on this topic show promise. Given that biomarkers are numerous, it leaves many opportunities for further discovery. In general, the advantages of biomarkers include non-invasive detection and prognostication of disease, the ability to detect a particular disease or disease state at an earlier timepoint before overt clinical manifestation, as well as the ability to predict disease course and treatment response. On the other hand, biomarkers may pale in comparison to traditional investigative tests in terms of their lack of sensitivity and/or specificity, which may create uncertainty. While many studies have attempted to identify the utility of biomarkers from many different sources of specimens, there still exist contradictions and disagreements between various studies and many opportunities for more investigations to uncover the unexplored areas of using biomarkers in CRS. Looking forward, with greater knowledge on how we may harness the benefits of biomarkers in the clinical management of patients with CRS, such patients can hopefully have their treatment plan optimized and tailored to their inflammatory endotype, potentially resulting in improved patient outcomes and satisfaction, as well as reduced treatment costs. 

## Figures and Tables

**Figure 1 diagnostics-13-00715-f001:**
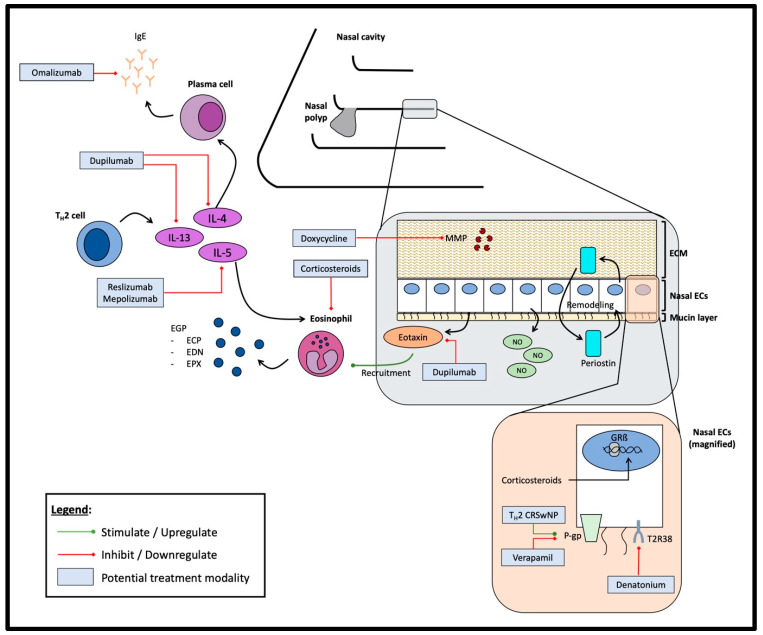
Interactions between various biomarkers in the pathomechanistic processes of different CRS subtypes, with various potential treatment modalities. IgE: Immunoglobulin E; EGP: Eosinophil granule protein; ECP: Eosinophilic cationic protein; EDN: Eosinophil-derived neurotoxin; EPX: Eosinophil peroxidase; MMP: Matrix metalloproteinase; ECM: Extracellular matrix; EC: Epithelial cells; P-gp: P-glycoprotein; NO: Nitric oxide; IL: Interleukin; GRß: Glucocorticoid receptor ß; T2R38: Taste receptor 2 member 38.

**Table 1 diagnostics-13-00715-t001:** Summarized biomarkers characterized by location found, utility, and CRS subtype. * Studies performed in murine model only. ^†^ Studies involving use of biologic therapy (e.g., monoclonal antibodies).

	Diagnosis	Prognosis	Treatment
**Eosinophil count**			
Blood	EosCRSwNP [11], EosCRS [12]	EosCRS [15], CRS [16]	
Tissue	CRS [22,23]	CRSwNP [18,20,21,22,24], CRS [22,23,25], EosCRSwNP [26]	
**Immunoglobulin E (IgE)**			
Serum		CRS [27], EosCRS [29]	CRSwNP [30] ^†^
Tissue	EosCRS [31]	CRSwNP [32,33,34,35]	CRSwNP [30] ^†^
Nasal secretions		CRSwNP [38]	CRSwNP [39] ^†^
**Eosinophil granule proteins (EGPs)**			
**Eosinophilic cationic protein (ECP)**			
Serum	CRSwNP [44], EosCRS [45]	CRS [46]	
Nasal secretions		CRS [47]	
**Eosinophil-derived neurotoxin (EDN)**			
Serum	EosCRS [45]		
**Eosinophil peroxidase (EPX)**			
Tissue	CRS [52]		
**Periostin**			
Serum	CRSwNP [55,56,57], EosCRSwNP [58]	EosCRS [55,59], CRSwNP [56]	
Tissue		CRS [60], EosCRSwNP [61], CRSwNP [55]	
**Interleukins (ILs)**			
**IL-5**			
Tissue	EosCRS [31,63,64], CRS [65]	CRSwNP [32,35,66], EosCRS [64]	
Nasal secretions		CRS [67]	CRSwNP [70,71] ^†^, CRS [65] ^†^
**IL-5/13**			
Nasal secretions	CRS [75], CRSwNP [75]		
**IL-4/13**			
NA			CRSwNP [76] ^†^
**IL-16**			
Tissue	EosCRS [77]		
**IL-25**			
Tissue	CRSwNP [78]	CRSwNP [78,79]	CRSwNP * [81] ^†^
**IL-33**			
Tissue	CRSwNP [82,84]	CRSwNP [83,84]	CRSwNP * [84] ^†^
**Oncostatin M**			
Tissue	CRSwNP [86]		
**Eotaxin**			
Plasma	EosCRS [89]		CRSwNP [39] ^†^
Tissue	CRSwNP [34], EosCRS [88]		
**Nitric oxide (NO)**			
Nasal exhalation	CRSwNP [92,93,96], CRSsNP [94], CRS [95]	CRSwNP [92,93,95,96,97]	CRS [93,97,98,99]
**Matrix metalloproteinase (MMP)**			
Tissue	CRSwNP [101,102,103,104]	CRS [105]	
Nasal secretions		CRS [106]	CRSwNP [107], CRS [108]
**P-glycoprotein (P-gp)**			
Tissue	CRSwNP [110,111]		
Nasal secretions	CRSwNP [112], CRS [112]		CRSwNP [113]
**Glucocorticoid receptor ß (GRß)**			
Tissue			CRSwNP [114,116]
**Mucin**			
Tissue			CRSwNP [119,120]
**Taste receptors**			
NA		CRS [121,122]	CRSsNP [123]

**Legend:** Blood/Serum/Plasma biomarkers: highlighted in orange. Tissue biomarkers: highlighted in green. Nasal secretions/exhalation product biomarkers: highlighted in blue. Existing knowledge gaps in literature: highlighted in grey.

## Data Availability

Not applicable.

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
