# Peer review of "The Diagnostic and Prognostic Role of Biomarkers in Chronic Rhinosinusitis"

_diagnostics, 2023, doi:10.3390/diagnostics13040715_

Round 1

Reviewer 1 Report

The authors propose a narrative review on all the possible biomarkers in CRS. The topic is interesting and important, even because today we still lack of clear evidences on them. 

I 've only minor issues: 

Chapter2.1: Eosinophil count: I think is fundamental at list brief paragraph on the new biological therapies, targeted against type 2 CRSwNP, in which a systemic and local eosinophilia is an important issue. lately the authors cite these therapies in IL section, but I'll suggest to introduce the topic here or in the introduction, to underline how often the eosinophilia is the first sign of a type 2 CRSwNP, that is further confirmed with a total IgE count

Better use prognosis instead os progmnostication in table 1

Author Response

Dear Reviewer,

Thank you for your valuable comments.

As per your suggestion, we have reorganised the introduction chapter, and added a new paragraph introducing the topic on biologic therapies (eg. monoclonal antibodies). We have also emphasized the importance of eosinophilia in type 2 inflammation in CRSwNP.

We have also altered the phrasing in Table 1 from “prognostication” to “prognosis” as per your suggestion.

We are grateful for your critique that leads to improvement of our paper.

Thank you.

Reviewer 2 Report

Dear Authors,

First of all congratulations for your work, which is so necessary for the ENT physician and not only. 

The manuscript is concise, focused on the CRS biomarkers and short, but the therapeutic implications are not well described. I suggest you include the role of  monoclonal antibodies, with the existing practical examples, in the Interleukins chapter and also in the Table 1, under the Treatment column. 

Author Response

Dear Reviewer,

Thank you for your valuable comments.

As per your suggestion, we have emphasized the role of monoclonal antibodies in CRS and its various targets in CRS, especially in the realm of type 2 inflammation (in which the current literature is mainly focused towards). We have also highlighted the various novel treatment modalities that involve monoclonal antibodies in Table 1 as per your suggestion, which serve as examples on how monoclonal antibodies can be potentially utilized in clinical practice given the current data available. We have also added a paragraph under the “Future Needs” section to elaborate further on the potential of monoclonal antibodies and how biomarkers in CRS can help support this endeavour.

We hope that we have satisfactorily addressed your comments, and are very grateful for your critique that leads to improvement of our paper.

Thank you.